# The Paradoxical Role of Uric Acid in Osteoporosis

**DOI:** 10.3390/nu11092111

**Published:** 2019-09-05

**Authors:** Kun-Mo Lin, Chien-Lin Lu, Kuo-Chin Hung, Pei-Chen Wu, Chi-Feng Pan, Chih-Jen Wu, Ren-Si Syu, Jin-Shuen Chen, Po-Jen Hsiao, Kuo-Cheng Lu

**Affiliations:** 1Division of Nephrology, Department of Medicine, Mackay Memorial Hospital, Taipei 10449, Taiwan (K.-M.L.) (P.-C.W.) (C.-F.P.) (C.-J.W.); 2Division of Nephrology, Department of Medicine, Fu Jen Catholic University Hospital, School of Medicine, Fu Jen Catholic University, New Taipei City 24352 Taiwan; 3Division of Nephrology, Department of Medicine, Min-Sheng General Hospital, Taoyuan City 33044, Taiwan (K.-C.H.) (R.-S.S.); 4Department of Medical Education and Research, Kaohsiung Veterans General Hospital, Kaohsiung 81362, Taiwan; 5Division of Nephrology, Department of Internal Medicine, Taoyuan Armed Forces General Hospital & Tri-Service General Hospital, National Defense Medical Center, Taipei 32551, Taiwan; 6Department of Life Sciences, National Central University, Taoyuan City 32001, Taiwan

**Keywords:** uric acid, osteoporosis, oxidative stress, inflammatory cytokines, vitamin D deficiency, secondary hyperparathyroidism

## Abstract

Because of its high prevalence worldwide, osteoporosis is considered a serious public health concern. Many known risk factors for developing osteoporosis have been identified and are crucial if planning health care needs. Recently, an association between uric acid (UA) and bone fractures had been explored. Extracellular UA exhibits antioxidant properties by effectively scavenging free radicals in human plasma, but this benefit might be disturbed by the hydrophobic lipid layer of the cell membrane. In contrast, intracellular free oxygen radicals are produced during UA degradation, and superoxide is further enhanced by interacting with NADPH oxidase. This intracellular oxidative stress, together with inflammatory cytokines induced by UA, stimulates osteoclast bone resorption and inhibits osteoblast bone formation. UA also inhibits vitamin D production and thereby results in hyper-parathyroidism, which causes less UA excretion in the intestines and renal proximal tubules by inhibiting the urate transporter ATP-binding cassette subfamily G member 2 (ABCG2). At normal or high levels, UA is associated with a reduction in bone mineral density and protects against bone fracture. However, in hyperuricemia or gout arthritis, UA increases bone fracture risk because oxidative stress and inflammatory cytokines can increase bone resorption and decrease bone formation. Vitamin D deficiency, and consequent secondary hyperparathyroidism, can further increase bone resorption and aggravated bone loss in UA-induced osteoporosis.

## 1. Introduction

Purines are found mostly in meat and internal organs such as the liver and kidneys, and its metabolism, mainly in liver, leads to the production of a waste product called uric acid (UA). Xanthine oxidase (XO) belongs to xanthine oxidoreductase, which catalyzes the oxidation of hypoxanthine into xanthine and, following, xanthine into UA. UA is the final oxidation product of purine catabolism in humans and higher primates, but it can be further oxidized into allantoin in most other mammal species. On average, the uric level in adult men is higher than that of women of similar age, which is due to different UA metabolisms in different sexes [1]. The mean value of UA in women is 2.5 to 7.5 mg/dL and in men 4.0 to 8.5 mg/dL [2].

Hyperuricemia is defined by increasing the serum UA level by 7.0 mg/dL in men and 5.7 mg/dL in women [3], which might result from the amplified production of UA or diminished UA excretion. Asymptomatic hyperuricemia is a term used to describe elevated UA levels that is accompanied with neither symptoms nor signs of monosodium urate (MSU) crystal deposition disease, such as gout arthritis or nephrolithiasis [4,5]. Urate-lowering therapy may be started until urate precipitates in urine sediment or there is urate articular damage examined by musculoskeletal ultrasound in asymptomatic hyperuricemia, because the benefit of urate-lowering therapy in asymptomatic hyperuricemia has not yet been proven regarding cardiovascular comorbidity or chronic kidney disease progression [6].

It is known that osteoporosis is a “metabolic” disease characterized by low bone mass and microarchitectural deterioration of bone tissue, leading to enhanced bone fragility and increased risk of bone fracture. Osteoporosis is usually considered to be a bone disease. However, it has been recently shown that this pathology involves the entire musculoskeletal system, and it is strongly coupled with alterations of fat metabolism and, therefore, of fatty acids [7,8,9,10]. Bone remodeling is a necessary process to repair damaged bone and maintain mineral bone hemostasis. The bone remodeling unit is a group of cells that continuously adjusts the microarchitecture of bone by osteoclasts and osteoblasts. The coupling of osteoclasts and osteoblasts is tightly controlled by multiple coupling factors, such as receptor activator of nuclear factor kappa-Β ligand (RANKL) and osteoprotegerin (OPG), wingless/integrated (Wnt) signaling, and semaphorin ligands. While coupling is deranged, the balance between bone resorption and bone formation is lost and results in bone loss [11,12]. The relationship between hyperuricemia and gout in developing osteoporosis is unclear. Thus, here we integrate the results of current studies and try to explain how they interplay.

## 2. The Clearance of Uric Acid (UA) in Humans

Kidneys are responsible for approximately two-thirds of UA elimination, while the intestine is responsible for one-third UA elimination [13]. Almost all UA is filtered from glomeruli; therefore, the extent of post-glomerular renal tubule UA reabsorption and secretion determine the final UA excreted in urine. The proximal renal tubule accounts for 90% of UA reabsorption, mainly the S1 segment. In the S2 segment of the proximal renal tubule, the amount of UA secretion is greater than its reabsorption. More distal sites of reabsorption in the proximal renal tubule are responsible for the remaining 10% of filtered UA [14]. There are three types of UA transporters in the regulation of uric acid levels in blood. Serum UA levels are mainly determined by urate transporters, which regulate the urate exchange in renal tubules, and include the following: two reabsorptive transporters—urate/organic anion exchanger (URAT1) and GLUT9—and two secretory transporters—members of the organic acid transporter (OAT) family and ATP-binding cassette subfamily G member 2 (ABCG2) (also called the breast cancer resistance protein or BCRP) [13,15]—where ABCG2 is also expressed in the intestinal epithelium. Among these exporter dysfunctions, ABCG2 is the most relevant genetic factor in the pathogenesis of hyperuricemia and gout patients based on genome-wide association study (GWAS). In hyperuricemic Japanese male, ABCG2 dysfunction decreased intestine urate excretion and, thus, increase the risk of renal overload [16]. In another association analysis in a hyperuricemic Japanese male, ABCG2 dysfunction caused significant underexcretion of renal urate [17,18]. Therefore, ABCG2 dysfunction contributes to the pathogenesis in both renal overload hyperuricemia and renal underexcretion hyperuricemia. Furthermore, loss-of-function mutation in *ABCG2 rs2231142* (*Q141K*) is associated with an increased risk of hyperuricemia and gout by decreased renal proximal tubule transport [15]. In addition, *ABCG2 rs2231142* is also significantly related to the poor response to allopurinol [19]. In the Taiwanese population, the *ABCG2 rs2231142-A* allele had a higher frequency of hyperuricemia in males or obese individuals [20].

## 3. Osteoporosis

### 3.1. Normal Bone Remodeling

Bone remodeling is a lifelong process wherein old bone is removed from the skeleton by bone resorption and replaced with new bone by bone formation. The bone remodeling unit comprises the osteoclast (OC), osteoblast (OB), and osteocytes. The bone remodeling process can be divided into six phases: quiescent, activation, resorption, reversal, formation, and mineralization [21]. In normal conditions, the length of the bone remodeling process is about 200 days, and the remodeling cycle is longer in cancellous bone than in cortical bone. The duration of bone remodeling is shortened in hyperthyroidism and hyper-parathyroidism, and it is longer in low bone turnover diseases like adynamic bone disease or osteomalacia [22].

Osteoblastic cells regulate osteoclast differentiation and activation through receptor activator of nuclear factor kappa-Β ligand (RANKL)-mediated signaling pathways. RANKL expressed in osteoblasts can be induced by 1,25 vitamin D, parathyroid hormone (PTH), and IL-6. The RANKL receptor is called RANK and is expressed in osteoclast progenitors to promote osteoclast differentiation and maturation. Macrophage colony-stimulating factor (M-CSF) is another important element to promote osteoclast proliferation and differentiation, and it is constitutively expressed in osteoblasts. Osteoprotegerin (OPG) is an inhibitory factor that inhibits osteoclast differentiation and activation by interfering with the binding of RANKL to RANK [23].

### 3.2. Coupling of Bone Stimulators and Turnover Inhibitors

There is increasing evidence to support the important role of ephrin and ephrin receptor (Eph) bi-directional signaling in normal bone coupling of bone resorption to bone formation [24]. Cells in the osteoclast lineage produce various coupling stimulators and inhibitors during the bone resorption process, which act on osteoblasts or their progenitors. The bone matrix also secretes transforming growth factor β (TGF-β) and insulin-like growth factor-I (IGF-I) during osteoclastic bone resorption and, subsequently, activates osteoblast bone formation. Several other coupling factors, such as semaphorin 4D, cardiotrophin-1, sphingosine-1-phosphate, bone morphogenetic protein 6, and Wnt 10b, have also been identified in bone unit communication [25].

### 3.3. The Pathogenesis of Osteoporosis

Several possible mechanisms of osteoporosis have been proposed, including:(a)Increased inflammatory cytokine-associated osteolysis. It leads to excessive activity of osteoclasts and leads to more bone resorption than bone formation. This phenomenon can be seen in gouty arthritis, inflammation, and vitamin D deficiency [26].(b)Incoordination of the RANK/RANKL system. Increased RANK/RANKL signal in OC strengthens the performance of osteoclasts and brings about more bone resorption. There are many clinical scenarios such as in hyperparathyroidism and some autoimmune diseases such as rheumatoid arthritis, systemic lupus erythematosus, and estrogen deprivation (postmenopausal women) [26,27].(c)Excessive Wnt signaling inhibitors in osteoblasts. Dickkopf-1, sclerostin, and secreted frizzled related proteins lead to diminished functioning of osteoblasts via a decrease in Wnt signaling activity, then reduced bone formation. This has been proven in chronic kidney disease, glucocorticoid-induced osteoporosis, and vascular calcification-related bone loss [28].

## 4. The Uric Acid Oxidant and Antioxidant Paradox

### 4.1. Antioxidant Properties of Uric Acid in Human Plasma

In human plasma, circulating UA acts as an antioxidant in several mechanisms (Figure 1). UA reacts with different oxidants including superoxide anions, hydrogen radicals, and, at the highest affinity, peroxynitrite. Peroxynitrite is a potent oxidant, generated by the rapid combination of free radical nitric oxide (NO) and superoxide, which can induce the inflammation response, lipid peroxidation, and tyrosine nitration [29,30]. Peroxynitrite also acts as an oxidant of tetrahydrobiopterin and leads to the uncoupling of nitric oxide synthase (NOS) [31]. Hence, peroxynitrite can increase superoxide and decrease NO production by eNOS uncoupling, and UA has protective effects against it. Besides, UA is an effective scavenger for peroxyl radicals (ROO^−^). Compared to ascorbic acid, UA is the major important water-soluble antioxidant in human plasma. Plasma UA levels are higher than plasma ascorbic acid levels, and UA has a higher reduction potential that leads to less iron and copper production, which is important for the Fenton reaction and hydroxyl radical generation [32,33]. Additionally, UA is an iron chelator to reduce iron-catalyzed oxidative stress reaction [34,35,36]. UA at physiologic concentrations has the ability to scavenge reactive oxygen species (ROS) and protect the erythrocyte membrane from lipid oxidation and further hemolysis [37]. Acute elevation of UA has a protective effect on cultured hippocampal neurons after ischemic insult and suppress the accumulation of reactive oxygen species after excitotoxic and metabolic insults [38]. Moreover, administration of UA can significantly attenuate the formation of nitrotyrosine in liver injury by hemorrhagic shock [39]. At the same time, UA can reduce neutrophil infiltration, which suggests that UA can prevent proinflammatory cell activation by oxidant stress [39]. Treatment with UA at a physiologic dose led to greater functional performance of the heart damaged by radicals and oxidants [40].

### 4.2. Intracellular Uric Acid Acts as a Pro-Oxidant to Damage Tissue

As shown in Figure 1, xanthine oxidase (XO) is the enzyme that catalyzes the oxidation of hypoxanthine to xanthine and of xanthine to UA, and there are two superoxide radicals produced during this process. Nicotinamide adenine dinucleotide phosphate oxidase (NADPH oxidase 4; NOX4) belongs to the NOX catalytic enzyme family that generates ROS. NADPH oxidase is located on the cell membrane and functions in intracellular electron transfer across the cell membrane, which converts extracellular molecular oxygen into superoxide radicals. Extracellular superoxide can dismutate to hydrogen peroxide. Both extracellular superoxide radicals and hydrogen peroxide can penetrate the cell membrane through chloride channel-3 (ClC3) and aquaporin channels, respectively, and subsequently initiate intracellular signaling and oxidative stress [41,42].

In addition, free radicals are produced during the process of UA degradation inside the cell. For example, an unpaired electron of the urate anion free radical is located on the five-membered ring of purine [43], and a carbon-centered free radical forms during the reaction of peroxynitrite to purine [31]. The reaction between UA and peroxynitrite is on a second-order rate and needs to consume oxygen. The final products of urate and peroxynitrite are allantoin, alloxan, and aminocarbonyl radicals. Allantoin and alloxan are produced during urate oxidation by other oxidants, whereas the aminocarbonyl radical is produced by peroxynitrite oxidation [44]. At physiological concentrations of UA, around 0.5 nM in human plasma, UA can increase the oxidation of liposomes and LDL promoted by the peroxynitrite-attacked end product, especially the aminocarbonyl radical [44]. It is worth to note that UA can react with peroxynitrite to produce radicals, and these radicals can further react with peroxynitrite to produce radicals in chain reaction. Furthermore, the antioxidant property of extracellular uric acid would be hindered by the hydrophobic conditions created by the lipid layer of the cell membrane, and oxidized lipids can convert uric acid into an oxidant with the help of copper [45,46]. The shift of UA between oxidant and antioxidant is related to the availability of lipid hydroperoxides formed during the early phase of LDL oxidation [46]. In differentiated mouse adipocytes and human subcutaneous primary adipocytes, UA significantly increases ROS production [47]. UA can also increase NAPHD oxidase activity directly and enhance intracellular superoxide generation. Production of peroxynitrite contributes to an inflammatory response that damages tissues by inducing lipid peroxidation. Both superoxide and peroxynitrite induce osteoclast detachment and exert osteoclast inhibition [48]. After interferon-gamma induction, nitrotyrosine production of human trabecular bone osteoblast increases, which means peroxynitrite is generated after inflammatory cytokines are stimulated in human osteoblast cells. The addition of peroxynitrite decreases the differentiation and proliferation of human trabecular bone osteoblasts [49]. All these findings suggest that high intracellular UA levels represent high inflammatory and oxidative stress, which contributes to the development of bone loss by disturbing osteoclast and osteoblast activities.

## 5. Pathophysiology of Uric Acid Induced Bone Loss

The imbalance between oxidative stress and antioxidation affects bone remodeling and causes osteoporosis. Oxidative stress inhibits MC3T3-E1 preosteoblast cell line differentiation and its mineralization ability [50]. Bai et al. demonstrated that oxidative stress inhibited osteoblast cell differentiation from rabbit bone marrow stromal cells via ERK and ERK-dependent NF-kB activation [51]. It is interesting to mention that increasing the intracellular oxidative stress by treatment with hydrogen peroxide or xanthine oxidase enhanced vascular osteoblast cell differentiation, but it decreased that of bone osteoblast cell differentiation. The above effect can be reversed by the usage of antioxidants, trolox and pyrrolidine dithiocarbamate (PDTC) [52].

It has been proven that osteoclasts contain NADPH oxidase, and in situ production of superoxide is responsible for bone resorption activity [53]. Superoxide has also been localized between the interface of osteoclast and ruffled border space by transmission electron microscopy [54]. Superoxide generated from xanthine and xanthine oxidase systems increase new osteoclast generation and bone resorption activities in calvarial bone organ culture and imply that adjacent free oxygen radicals generated in bone can promote osteoclast formation [55]. Moreover, hydrogen peroxide also promotes osteoclast formation in a concentration-dependent manner and, thereby, increases bone resorption [56,57].

During acute gout arthritis attacks, crystallization of needle-shaped monosodium urate (MSU) and infiltration of neutrophils are characteristics in the pathogenesis in synovial fluid. After phagocytosis of MSU by human monocyte-induced NLRP3 inflammasomes and release of proinflammatory cytokines, such as IL-1 [58], TNF-α [59], IL-6 [60], and IL-8 [61], inflammatory cells are thereby recruited, mainly neutrophils, into the joint space and initiate gouty inflammation attacks. Thioredoxin-interacting protein (TXNIP) is a protein that links oxidative stress to inflammasome activation [62], and the primary function of TXNIP is inhibition of thioredoxin, which functions as a redox protein. While MSU induces TXNIP expression upon ROS activation, it causes a dissociation of TXNIP from oxidized thioredoxin and allows TXNIP bind to NLPR3. Once NLPR3 is activated, caspase-1 activity is increased and then generates IL-1β [63]. This pathway can be blocked by allopurinol, which alleviates ROS levels.

The generation of inflammatory cytokines is highly associated with bone cell activity. Interleukin-1 (IL-1) acts as an important activator in bone resorption in cooperation with receptor activator of nuclear factor kappa-B ligand (RANKL) signaling to promote osteoclast progenitor proliferation and differentiation. IL-1 produced from osteoblasts can induce more RANKL expressions in osteoblasts, while IL-1 produced from bone marrow cells can promote osteoclast progenitors to form osteoclasts [64]. IL-6 has a positive role in osteoclast differentiation, indirectly, by increasing RANK expression in osteoblasts and, thereby, activating the RANKL downstream pathway in osteoclasts, including nuclear factor kappa-light-chain-enhancer of activated B cells (NF-κB) JNK and p38. While the expression of soluble IL-6 receptors on osteoblasts is up-regulated in certain conditions such as inflammation, IL-6 increases RANKL expression in osteoblasts and, thereby, increases osteoclast bone resorption [65]. Tumor necrosis factor-α (TNF-α) is a key element in the pathogenesis of inflammation, which promotes RANKL expression in macrophages, bone marrow stromal cells, and macrophage colony-stimulating factor (M-CSF) on stromal cells. Both RANKL and M-CSF production is the mechanism by which TNF-α stimulates osteoclast differentiation and activity in inflammatory arthritis [66]. The bone resorption of osteoclasts induced by TNF-α can be further promoted in the presence of IL-1, and both are important elements in inflammatory bone disease [67].

Both inflammatory cytokines and oxidative stress stimulate osteoclast activity and inhibit osteoblast function, which result in bone loss (Figure 2). Allopurinol and oxypurinol are both used to treat hyperuricemia and gout by inhibiting XO activity and, consequently, reducing the conversion of xanthine to UA. Allopurinol and oxypurinol increase osteoblast differentiation and, following, increase bone formation [68]. However, allopurinol and oxypurinol have no effect on osteoclast formation and activity [68]. XO inhibition by allopurinol and oxypurinol reduce serum UA and, thereby, rescue the inhibitory effect on UA osteoblasts. Also, XO inhibitors can reduce ROS production during UA metabolism, which is harmful for osteoblast differentiation and bone mineralization.

## 6. The Impact of Uric Acid on Vitamin D Metabolism

Several studies have emphasized the impact of UA on vitamin D metabolism. UA can directly inhibit 1α-hydroxylase expression in renal proximal tubules and thereby reduce 1,25(OH) vitamin D (1,25D) concentrations in hyperuricemic rats. This phenomenon can be reversed with febuxostat treatment [69]. In the same study, the expression 24-hydroxylase, which is used to degrade 25D or 1,25D, is enhanced in hyperuricemic rats [69]. After receiving allopurinol to lower serum UA levels, serum 1,25D levels rise more than 20% in patients with chronic renal failure [70]. Among postmenopausal Chinese Han women, hyperuricemia is significantly associated with vitamin D insufficiency (VDD, 25D < 30 ng/mL), and 25D levels in the lowest quartile had higher serum UA levels than those in the highest quartile group [71]. Likewise, the serum UA level is negatively associated with vitamin D levels in chronic kidney disease (CKD) stage 3a to 5 patients [72]. The association between VDD and hyperuricemia can be postulated by UA inhibition of hepatic 25-hydroxylation or PTH-induced ABCG2 down-regulation in the intestine and proximal renal tubules, which results in hyperuricemia.

On the other hand, the existence of a relationship between UA and PTH level are also observed. Chen et al. concluded from the data of the National Health and Nutrition Examination Survey (NHANES) that serum UA was positively correlated with elevated parathyroid hormone (PTH) levels, especially in at an estimated glomerular filtration rate (GFR) <60 mL/min/1.73 m^2^ [69]. The increase of UA is in parallel with the increase of PTH, and PTH in the highest quartile had significantly higher serum UA than those in the lowest quartile [73]. Yoneda et al. demonstrated serum UA was significantly higher in patients with primary hyperparathyroidism (PHP) and in recovery after parathyroid adenoma removal. They proposed PTH might affect UA metabolism and reduce UA clearance [74]. In a cross-sectional study, the serum PTH level was positively correlated with serum UA level, and the PTH level was significantly higher in patients with hyperuricemia than normouricemia [75]. After receiving a parathyroidectomy for PHP, serum UA levels decreased in postoperative follow up [76,77]. Furthermore, teriparatide is a recombinant PTH and used as anabolic agent in treating osteoporosis. Usage of teriparatide in postmenopausal women increased the incidence of hyperuricemia episodes in a dose-dependent manner, although the attack of gout did not reach significance [78]. In fact, the mechanism by which PTH increases UA remains unclear. Hisatome et al. reported that reduced uric acid transport in proximal renal tubules might cause hyperuricemia in PHP patients [79]. Ryusei et al. reported ABCG2 expression in renal proximal tubules and intestines are both downregulated, and this resulted in decreased excretion of UA via kidneys and intestines. The association between PTH and UA is further supported by the use of cinacalcet in secondary hyperparathyroidism (SHPT). Cinacalcet is an allosteric activator of the calcium-sensing receptor and acts as a novel therapy to suppress PTH level in SHPT. Cinacalcet can significantly reduce the increase of serum UA levels in SHPT without affecting renal function [80]. In the same study, after 12 weeks of cinacalcet treatment, serum UA level decreased together with the reduction of PTH in patients with SHPT undergoing dialysis [80]. It is important to mention that the effect of increased PTH by serum UA was augmented by inadequate vitamin D levels because VDD can stimulate more PTH production in the parathyroid gland. Hyperuricemia-induced VDD and hyper-parathyroidism can further aggravate bone remodeling disturbances in UA-related bone loss and dramatically increase fracture risk.

## 7. The Association of Uric Acid and Bone Health in Epidemiological Studies

### 7.1. Uric Acid as a Protective Factor in Bone Loss under Normal High Levels

UA is the primary antioxidant in human plasma and accounts for more than 60% of the capacity to scavenge free oxidative radicals in serum [81]. Serum UA in the normal physiologic range acts as an antioxidant to prevent bone loss and osteoporosis. The Concord Health and Ageing in Men Project (CHAMP) is a large, cross-sectional study of older men, with a mean UA value of 6.1 ± 1.3 mg/dL. Reported serum UA levels more than 6.0 mg/dL are associated with greater bone mineral density (BMD) at all skeletal sites than serum UA less than 6.0 mg/dL, even after adjustment for age, body mass index, calcium, intact PTH, serum 25D concentration, and bisphosphonates [82]. Furthermore, an increase in serum UA level is also associated with a lower prevalence of osteoporosis. In the same study, serum UA was inversely correlated with urinary concentrations of the amino-terminal cross-linked telopeptide of collagen type 1, NTX-1 (a biomarker for osteoclasts, bone resorption), but not seen in amino-terminal procollagen type 1 propeptide, P1NP (a biomarker for osteoblasts, bone formation) [82].

Likewise, among 470 Japanese women, with a mean serum UA value of 4.7 ± 1.0 mg/dL, a higher serum UA level was significantly, positively associated with spine BMD [83]. Another study in peri- and postmenopausal women, with a serum UA value of 4.3 ± 1.0 mg/dL, women with a higher serum UA level had greater BMD at all skeletal sites at baseline and had a lower bone loss rate after 9.7 years of follow up [84]. The Osteoporotic Fractures in Men (MrOS) Study is a larger case–cohort, prospective study designed to understand the relation of serum UA in fracture risk among men more than 65 years old. This study demonstrated an 18% decrease of nonspine fractures per 1 SD increase of baseline serum UA, and hip BMD linearly increased with serum UA level, although the hip fracture rate was not associated with the serum UA level [85].

In brief, several studies have shown the benefit of UA in bone health, and UA at physiological concentrations is considered as a potent antioxidant factor to protect against oxidative stress associated bone loss and osteoporosis. UA is an effective scavenger for oxidative stresses such as superoxide, hydroxyl radicals, and, most potent, oxidant peroxynitrite. Oxidant stress has been identified to attenuate bone remodeling and is a risk factor for osteoporosis. Several antioxidant agents, like vitamin C [86,87], vitamin E [88,89,90], superoxide dismutase [91,92], and glutathione peroxidase [93], can also provide benefits in bone quality improvement and reduce fracture risk. Antioxidant properties of UA can also protect against free radical damage to vessels, the heart, and neurons [37,94,95].

### 7.2. Uric Acid as a Risk Factor in Bone Loss with Hyperuricemia and Gout

UA is linked to bone loss in hyperuricemia and gout. Several studies have demonstrated that UA plays an important role in the pathogenesis of osteoporosis. The Cardiovascular Health Study (CHS) is a population-based, longitudinal study of community-dwelling adult more than 65 years old, and it showed that serum UA was significantly associated with hip fractures in men, not in women. Hyperuricemia led to a 60% increased risk of hip fracture in men after adjustment of covariates, including BMI. If serum UA >8 mg/dL, the risk of hip fracture is dramatically elevated [96]. The Nurses’ Health Study enrolled 103,799 women with a long period of follow-up (wrist fracture: 14 years; and hip fracture: 22 years) and showed that the risk of hip fracture increased by 38% if there was a history of gout attack, but the risk with wrist fracture did not reach significance [97].

### 7.3. Potential Mechanism by which UA Disturbs Bone Remodeling

Both inflammatory cytokines and oxidative stress caused by MSU represent common pathophysiologies raised to explain the association of gout or hyperuricemia with fracture risk. Alteration of the bone remodeling unit by inflammatory cytokines and oxidative stress has been described in the previous section. In brief, both inflammatory cytokines and oxidative stress induce osteoclast activity and suppress osteoblast activity, which results in bone loss and osteoporosis. Another mechanism by which UA compromises bone heath is by VDD and hyper-parathyroidism in hyperuricemia or gout (Figure 3).In fact, vitamin D regulates osteoblast function and extracellular matrix mineralization, and it also promotes osteoclast differentiation and maturation through induction of RANKL expression in osteoblasts [26,98,99,100]. In VDD, the concentration of vitamin D is inversely related to the PTH level, and a lower 25D level significantly increases bone resorption and bone formation markers [101]. Collagen type 1 cross-linked C-telopeptide (CTX) is a surrogate biomarker for measuring the metabolic product of bone resorption. Serum CTX was significantly elevated in VDD in a German birth cohort [102]. However, the relationship between vitamin D concentration and bone resorption or formation biomarkers is still inconclusive because of different study group ages, ethnicities, or countries in various latitudes. A randomized, double-blind, placebo-controlled trial in postmenopausal women showed that supplementation with 1000 IU of vitamin D_3_ daily can significantly increase both serum CTX and P1NP levels, which means administration of vitamin D can restore the decrease in bone resorption and formation [103]. Additionally, the PTH level significantly decreases together with an elevated vitamin D level after vitamin D is given [103].

In the case of VDD, an elevated PTH level is observed because an active form of vitamin D, 1,25-hydroxyvitmain D (1,25D), has an ability to suppress PTH production in the parathyroid gland. This elevated PTH level in VDD is called secondary hyper-parathyroidism (SHPT). VDD and SHPT are both proposed as pathologies in bone fractures and can further aggravate bone loss in MSU-induced osteoporosis [104,105]. Ooms et al. reported in elderly women that, only with 25D levels below 30 ng/mL, the 25D level was negatively associated with serum PTH and the bone formation marker, osteocalcin, and the 25D level was positively associated with femoral neck and trochanter BMD. Moreover, serum PTH was negatively associated with BMD in femoral neck, trochanter, and distal radius bones. The study concluded that VDD contributed to bone fracture risk in postmenopausal women by increasing bone turnover and leading to bone loss [106]. Similar findings can also be seen in premenopausal women [107].

The AGES-Reykjavik Study in Iceland showed 25D levels less than 12 ng/mL were associated with a higher risk for hip fracture (HR 2.24, 95% CI: 1.63, 3.09) as compared to 20–30 ng/mL, but there was no significant difference between 12–20 ng/mL and >20 ng/mL. BMD in the femoral neck was significantly lower when 25D levels were less than 12 ng/mL than the control [108]. However, in the Health Aging and Body Composition Study conducted in Pittsburgh and Memphis, there was no association between 25D levels and nonspine fracture [109]. Actually, the association between vitamin D and bone fractures remains inconclusive. In order to explore the association between 25D and hip fractures, and what the optimal 25D concentration for bone health is, a meta-analysis of prospective cohort studies was conducted and demonstrated that the adjusted relative risk of lowest versus highest 25D levels in hip fractures was 1.58 (95% CI: 1.41, 1.77) [110]. Besides, 25D levels less than 24 ng/mL have an increased risk of hip fracture [110].

The association between gout attack and bone fracture is still in debate. In Taiwan, the nationwide, population-based, retrospective matched cohort study showed that the gout cohort had an increased risk of bone fractures in the spine and lower limbs, especially in female individuals. The XO inhibitor can reduce serum UA and, thereby, prevent gout arthritis. The risk of facture among gout patients was also significantly reduced (HR 0.72, 95% CI: 0.67, 0.78). Benzbromarone is used to increase urinary UA excretion, and it had a similar protective effect on facture risk (HR 0.71, 95% CI: 0.68, 0.75). Hence, UA-lowering therapy implies that gout history predisposes patients to bone loss, primarily due to high serum UA levels [111]. Another Taiwan study showed similar findings, that individuals with a history of gout had a modest increase in developing osteoporosis in the future [112].

In contrast to Taiwan studies, the Clinical Practice Research Datalink study in the United Kingdom revealed that gout attack was not associated with bone fracture, and UA-lowering therapy had no benefit on the long-term fracture risk [113]. Similarly, Basu et al. reported allopurinol cannot reduce hip fracture risk in elderly adults [114].

In the gray frames, UA or gout induces the circulation of cytokines and reactive oxygen species (ROS). A decrease in osteoblast cells (OB) and enhanced osteoclast cells (OC) were affected, which then leads to bone loss. In the blue frame, the UA inhibits 1α-hydroxylase (1α-OHase) activity and causes a decrease in 1,25-dihydroxyvitamin D3 (1, 25(OH)_2_D_3_) formation. Elevated parathyroid hormone (PTH) levels will caused a decrease in UA secretion from renal proximal convoluted tubules (PCTs) due to 1, 25(OH)_2_D_3_ insufficiency. Then, hyperuricemia increases after UA is accumulated. In the yellow frames, UA also enhances 24-hydroxylase (24-OHase) activity, which leads to decrease products of 25(OH)_2_D_3_ and 1, 25(OH)_2_D_3_. They caused a reduction in OB viability. In the purple frames, increased inflammatory cytokines and osteoblast cell receptor activators of nuclear factor kappa-B ligand (RANKL) would promote OC activity. Bone loss could occur under increased OC activity and decreased OB viability.

## 8. Magnetic Resonance Spectroscopy (MRS) for Osteoporosis

It is well established that levels of marrow fat are higher in older adults with osteoporosis, defined by either low bone density or vertebral fracture. MRS and positron emission tomography allow noninvasive quantification of bone marrow physiology and pathology on a large scale. Specifically, MRS can monitor metabolic paths in vivo to clarify the development of osteoporosis. It is associated with increased marrow fat accompanied by a preferential increase in unsaturated lipids [9]. Since MRS of bone marrow lipid profiles from peripheral skeletal sites may be a promising tool for screening large populations to identify individuals with or at risk for developing osteoporosis [10]. There is also accumulating evidence that points to the potential for bone marrow fat (BMF) to play diagnostic and therapeutic roles in the prevention of osteoporosis [115,116]. A better understanding of the role of BMF in humans may lead to new avenues to promote bone formation and, thus, prevent and treat osteoporosis. Future developments in the clinical use of MRS to quantify data in human blood and bone marrow could be promising.

## 9. Conclusions

To reduce the likelihood of high uric acid levels, patients should initially limit their consumption of certain purine-rich foods and avoid alcoholic drinks and beverages sweetened with high-fructose corn syrup. Extracellular uric acid (UA) has antioxidant properties by effectively scavenging free radicals in human plasma, but this benefit might be disturbed by the hydrophobic lipid layer of the cell membrane. In contrast, intracellular free oxygen radicals are produced during UA degradation, and superoxide is further enhanced by interacting with NADPH oxidase. This intracellular oxidative stress, together with inflammatory cytokines induced by UA, stimulate osteoclast bone resorption and inhibit osteoblast bone formation. UA also inhibits vitamin D production and thereby results in hyperparathyroidism, which causes less UA excretion in the intestine and renal proximal tubules by inhibiting the urate transporter ATP-binding cassette subfamily G member 2 (ABCG2). At normal or high levels, UA is associated with a reduction in bone mineral density and protects against bone fracture. However, in hyperuricemia or gout arthritis, UA increases the bone fracture risk because oxidative stress and inflammatory cytokines can increase bone resorption and decrease bone formation. Vitamin D deficiency and consequent secondary hyper-parathyroidism can further increase bone resorption and aggravate bone loss in UA-induced osteoporosis. 

## Figures and Tables

**Figure 1 nutrients-11-02111-f001:**
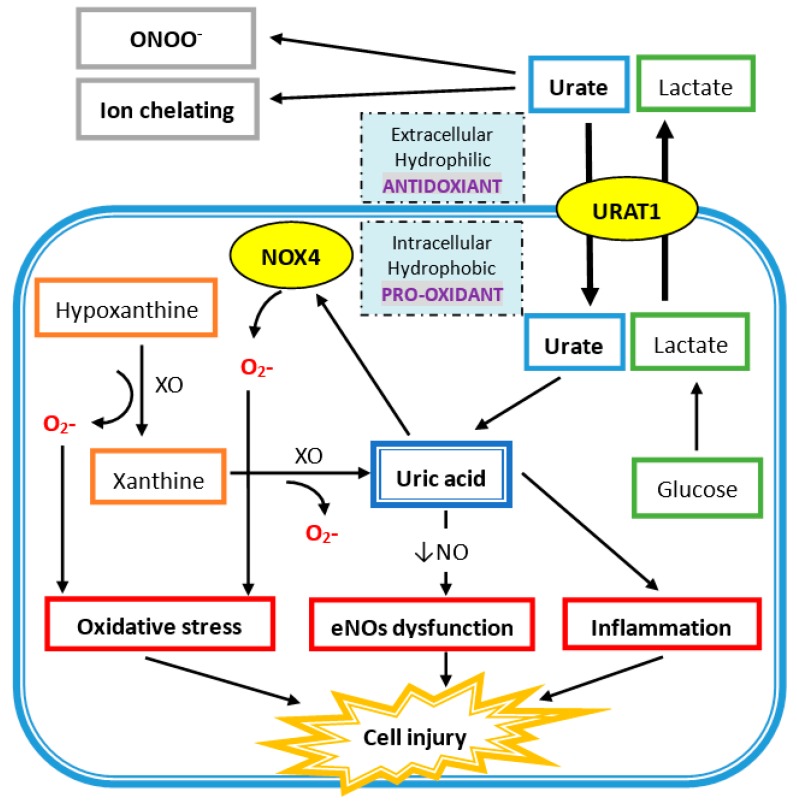
Uric acid (UA) acts as a hydrophobic pro-oxidant within the cell and a hydrophilic antioxidant outside the cell. Urate enters the cell via UA transporter 1 (URAT1) on the cell membrane. UA is produced from xanthine via xanthine oxidase (XO) in the cell and also generates superoxide ions (O_2_-), which also promotes oxidative stress by superoxide free radicals produced via NADPH oxidase (NOX4). UA intracellularly induces endothelial nitric oxide synthase (eNOs) with a decrease in nitric oxide (NO) generation. It also directly increases inflammation, which leads to cell injury.

**Figure 2 nutrients-11-02111-f002:**
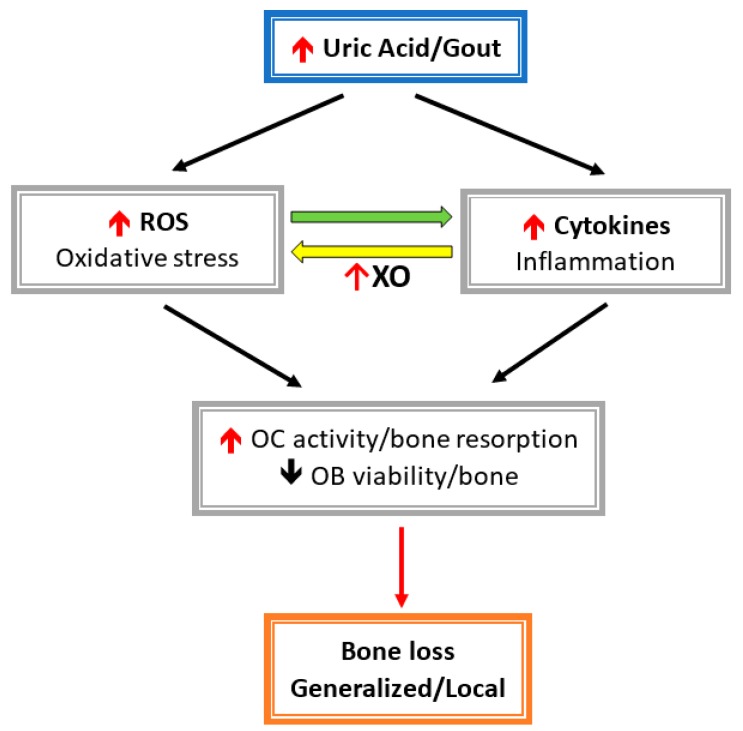
Hyperuricemia-/gout-induced inflammatory cytokines and related reactive oxygen species (ROS) activate osteoclast (OC) activity and inhabit osteoblast cell (OB) viability. Cytokines induce oxidative stress via xanthine oxidase (XO), then leads to enhanced bone loss.

**Figure 3 nutrients-11-02111-f003:**
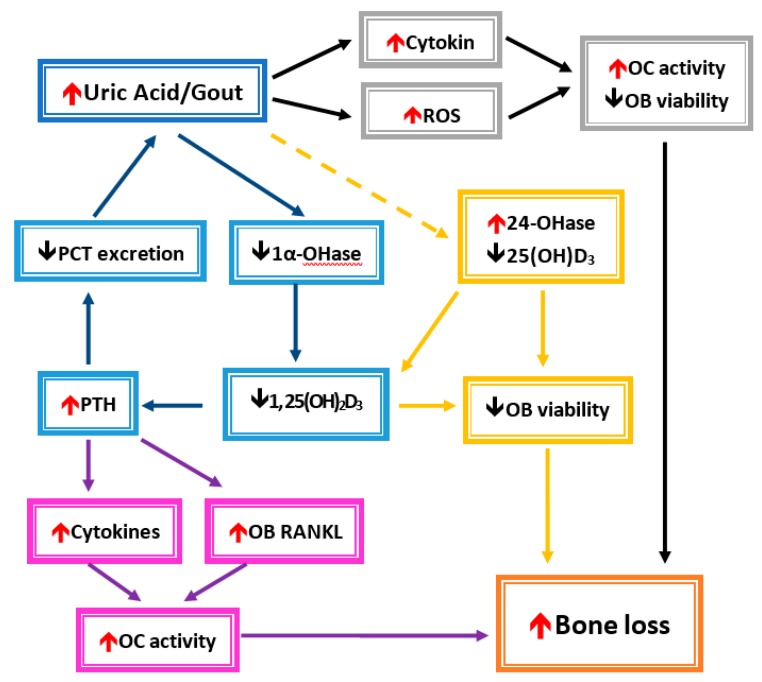
Conceivable mechanisms of hyperuricemia-/gout-induced bone loss.

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
