# Peer review of "The Paradoxical Role of Uric Acid in Osteoporosis"

_nutrients, 2019, doi:10.3390/nu11092111_

Round 1

Reviewer 1 Report

The manuscript addresses the role of uric acid (UA) in relation to the pathology of osteoporosis. The authors stress that depending on its concentration in the blood, UA can play an antioxidant role and therefore be protective for damage from osteoporosis or trigger a chain of oxidations involving fats and contribute to bone fragility.

The paper is technically sound and the subject is clearly explained by the authors. All the main papers are mentioned. However, I suggest minor changes.

The first change concerns the definition of osteoporosis.

The definition of osteoporosis reported by authors (page 2 lines 55-57) is obsolete. Nowadays it is known that osteoporosis is a “metabolic” disease characterized by low bone mass and micro-architectural deterioration of bone tissue, leading to enhanced bone fragility and increased risk of bone fracture. Osteoporosis is usually considered to be a disease of bone. However, it has been recently shown that this pathology involves the entire musculoskeletal system and it is strongly coupled with alterations of fats metabolism [1-4].

I believe that, in relation to the subject of the manuscript, the authors should modify and broaden the definition of osteoporosis, including the fact that it involves a modification of the quality of bone marrow and therefore of fatty acids [1-8].

A second suggestion is related to the expansion of readers potentially interested in the results reported by the authors in this manuscript. Magnetic resonance spectroscopy (MRS) and positron emission tomography allowing non-invasive quantification of bone marrow physiology and pathology on a large scale. Specifically MRS can in vivo monitor metabolic paths to clarify some still open problems related to osteoporosis developments. Therefore I suggest to discuss possible future development of the research using also MRS to quantify data in human blood and in bone marrow.

In conclusion, have authors sufficient data to conclude something about indication to lower uric acid levels? (as an example, limit purine-rich foods…)

Suggested REFERENCES

1.  WHO Study Group 2003 Prevention and management of osteoporosis. Report of a WHO Study Group WHO Tec Rep Ser 921 1-204.

2.  James Francis Griffith

Bone Marrow Changes in Osteoporosis In book: Osteoporosis and Bone Densitometry Measurements

        January 2013; DOI: 10.1007/174_2012_614

3. Yeung DKW, Griffith JF, Antonio GE et al. 2005. Osteoporosis is associated with increased marrow fat content and decreased marrow fat unsaturation: a proton MR spectroscopy study. J Magn Reson Imaging 2005;22 279-285.

4.   Di Pietro G, Capuani S, Manenti G. et al. Bone Marrow Lipid Profiles from Peripheral Skeleton as Potential Biomarkers for Osteoporosis: A 1H-MR Spectroscopy Study. Acad Radiol 2016; 23:273–283. 

5.  Schwartz AV. Marrow fat and bone: review of clinical findings. Front

       Endocrinol (Lausanne) 2015; 6:40.

6.   Bermeo S, Gunaratnam K, Duque G. Fat and bone interactions. Curr Osteop Report 2014; 12:235–242.

7.   Yeung DKW, Lam SL, Griffith JF, et al. Analysis of bone marrow fatty acid composition using high-resolution proton NMR spectroscopy. Chem Phys Lipids 2008; 151:103–  109.

8.  Ren J, Dimitrov I, Sherry AD, et al. Composition of adipose tissue and marrow fat in humans by 1H NMR at 7 tesla. J Lipid Res 2008; 49:2055–2062.

Reviewer 2 Report

Overall, this is a well-written paper on role  of uric acid in osteoporosis; the rationale and conclusion are adequate.  

It is well-known that high plasma UA level is strongly associated with gout and coronary vascular diseases but the relationship with osteroporosis has not been explained well so this paper is valuable.
